# Meta-Analysis of the Public RNA-Seq Data of the Western Honeybee *Apis mellifera* to Construct Reference Transcriptome Data

**DOI:** 10.3390/insects13100931

**Published:** 2022-10-14

**Authors:** Kakeru Yokoi, Takeshi Wakamiya, Hidemasa Bono

**Affiliations:** 1Insect Design Technology Module, Division of Insect Advanced Technology, Institute of Agrobiological Sciences, National Agriculture and Food Research Organization (NARO), 1-2 Owashi, Tsukuba 305-8634, Ibaraki, Japan; 2Laboratory of Genome Informatics, Graduate School of Integrated Sciences for Life, Hiroshima University, 3-10-23 Kagamiyama, Higashi-Hiroshima City 739-0046, Hiroshima, Japan; 3Laboratory of BioDX, Genome Editing Innovation Center, Hiroshima University, 3-10-23 Kagamiyama, Higashi-Hiroshima City 739-0046, Hiroshima, Japan

**Keywords:** *Apis mellifera*, honeybee, meta-analysis, public RNA sequencing data, reference transcriptome

## Abstract

**Simple Summary:**

The honeybee is an economically and ecologically essential insect species and a critical social behavior model. It is vital to honey production and pollination. Therefore, it has been studied extensively in entomological research. The Western honeybee (*Apis mellifera*) is a representative species of honeybees. Reference transcriptome data are required to perform the meta-analyses using public RNA-Seq data and transcriptome analysis using newly sequenced RNA-Seq data. The reference transcriptome sequences were constructed, which consisted of 149,685 transcripts, and 194,174 predicted amino acid sequences. Half the predicted protein sequences were functionally annotated using protein sequence datasets for several model and insect species. We performed meta-analysis to search for novel immune-response-related transcripts. We identified 3–20 transcripts that were upregulated or downregulated in response to both viral and bacterial infections. However, the immune-related transcripts were not included among them. Autophagy-related protein 3 was downregulated by viral infections. The results suggested that autophagy was implicated in the viral immune responses, and autophagy was not regulated by the canonical immune pathways in *A. mellifera*. The results of the meta-analysis suggested that the reference transcriptome data could be used for transcriptome analysis in *A. mellifera*.

**Abstract:**

The Western honeybee (*Apis mellifera*) is valuable in biological research and agriculture. Its genome sequence was published before those for other insect species. RNA-Seq data for *A. mellifera* have been applied in several recently published studies. Nevertheless, these data have not been prepared for use in subsequent meta-analyses. To promote *A. mellifera* transcriptome analysis, we constructed reference transcriptome data using the reference genome sequence and RNA-Seq data curated from about 1,000 runs of public databases. The new reference transcriptome data construct comprised 149,685 transcripts, and 194,174 protein sequences were predicted. Approximately 50–60% of the predicted protein sequences were functionally annotated using the protein sequence data for several model and insect species. Novel candidate immune-related transcripts were searched by meta-analysis using immune-response-related RNA-Seq and reference transcriptome data. Three to twenty candidate transcripts including *autophagy-related protein 3* were upregulated or downregulated in response to both viral and bacterial infections. The constructed reference transcriptome data may facilitate future transcriptome analyses of *A. mellifera*.

## 1. Introduction

The honeybee (Hymenoptera; Apidae; *Apis* spp.) is an economically and ecologically important insect species and a key social behavior model [1]. It is vital to honey production and pollinates both domesticated crops and wild plants. It has also been studied extensively as a model social insect species in entomological research. *Apis mellifera* (Western honeybee) is a representative *Apis* species that is widely applied in biological research, as a pollinator, and for honey production. In view of its utility and versatility, its reference genome sequence data were published in 2006 without the use of next-generation sequencing (NGS) [2]. The reference genome sequence data were continuously updated and chromosome-level genome assembly data were finally published in 2019 following advances in DNA sequencing technology [3]. Certain investigations using genomic and transcriptomic data for *A. mellifera* disclosed novel insights based on molecular and genetic evidence ([4]). Hence, the amount of *A. mellifera* transcriptome data generated by RNA sequencing (RNA-Seq) has increased in public databases.

The foregoing findings suggest that RNA-Seq and transcriptome data, as well as studies using the RNA-Seq data of *A. mellifera*, are accumulating. We previously found >80 transcriptomic expression datasets for *A. mellifera* in the integrated transcriptome database known as All of Gene Expression (AOE) [5]. We also discovered several studies using *A. mellifera* RNA-Seq published over the past two years [6,7,8,9]. Future studies should use detailed transcriptomic data including isoform sequence data for single genes, non-coding RNA sequences, and annotations of each isoform. We constructed chromosome-level genome assembly data and gene sets [10] and reference transcriptome data [11] for the domestic silkworm (*Bombyx mori*) and demonstrated that studies could be conducted utilizing reference transcriptome data [12,13]. Considering the foregoing results, reference transcriptomic data for *Apis mellifera* should be constructed to foster *A. mellifera* research using transcriptomic analysis.

Meta-analysis is a promising method to use and combine the data and observations of different studies to obtain novel biological insights. We previously performed a meta-analysis using RNA-Seq or genomic data derived from various studies and available in public databases. We used public RNA-Seq data related to hypoxia to identify new candidate genes induced by hypoxic stimulation [14]. New candidate genes related to oxidative stress and hypoxia were identified using public RNA-Seq data associated with these pathological processes. Several of the genes identified were annotated as cell cycle functions [15]. Landscapes of the transposable elements (TEs) in *Apis* species were constructed using public genome sequence data for multiple *Apis* species. Most of the *Apis* TEs identified belonged to only a few families [16]. In addition, Doublet et al., showed that genes or molecular pathways in *A. mellifera* which were responded to multiple pathogens were identified using transcriptome dataset of RNA-Seq, microarray, or tilling array from diverse tissues [17]. The results of earlier studies suggested that meta-analyses involving sequence data in public database are highly effective research methods.

In the present study, we used a meta-analysis to construct reference transcriptome sequences comprising *A. mellifera* RNA-Seq data and chromosome-level genome assembly [3] from public databases. We then functionally annotated the constructed transcriptome sequences. We quantified the transcriptome expression levels using the public RNA-Seq data to show the usefulness of the reference data. We extracted the sample RNA from *A. mellifera* subjected to several microbial pathogens and identified candidate genes involved in the host immune response (Figure 1).

## 2. Materials and Methods

### 2.1. RNA-Seq Data Retrieval from the Public Database

The NCBI Gene Expression Omnibus database (GEO, https://www.ncbi.nlm.nih.gov/geo/; accessed on 1 August 2022) was used to compile a list of *A. mellifera* RNA-Seq data. Each selected sequence read archive (SRA) file was downloaded using the ‘prefetch’ command in sra-tools v. 2.9.6 (https://github.com/ncbi/sra-tools; accessed on 1 August 2022) and converted to FASTQ file using the ‘fasterq-dump’ command. The FASTQ files were compressed and trimmed using the default options in Trim Galore! v. 0.6.6 (https://github.com/FelixKrueger/TrimGalore; accessed on 1 August 2022).

### 2.2. Constructing and Evaluating the Reference Transcriptome Sequence Data

The trimmed RNA-Seq reads were mapped to the *A. mellifera* genome (v. Amel_HAv_3.1; GenBank assembly accession ID No. GCA_003254395.2) [3] using HISAT2 v. 2.2.1 (https://daehwankimlab.github.io/hisat2/; accessed on 1 August 2022) [19]. The output SAM files were converted to BAM files with Samtools v. 1.12 (https://sourceforge.net/projects/samtools/files/samtools/1.12/; accessed on 1 August 2022) [20] and the mapped BAM file data were used to assemble the reference transcriptome with the reference *A. mellifera* gene model GCF_003254395.2_Amel_HAv3.1_genomic.gtf.gz (https://ftp.ncbi.nlm.nih.gov/genomes/all/annotation_releases/7460/104/GCF_003254395.2_Amel_HAv3.1/GCF_003254395.2_Amel_HAv3.1_genomic.gtf.gz in NCBI Refseq accessed on 1 August 2022) in a GTF file via StringTie v. 2.1.7 (https://github.com/gpertea/stringtie/releases; accessed on 1 August 2022) [19]. The reference transcriptome sequence data in the GTF file were constructed by merging the transcript data of each RNA-Seq via the ‘stringtie-merge’ command. The GTF file of the reference transcriptome was converted into a FASTA file with gffread v. 0.12.1 (https://github.com/gpertea/gffread/releases; accessed on 1 August 2022) [21]. BUSCO v. 5.2.2 was used for the evaluation of the quality of the constructed transcriptome sequences [22]. The amino acid sequences of the reference transcriptome were predicted with TransDecorder v. 5.5.0 (https://github.com/TransDecoder/TransDecoder/wiki; accessed 1 August 2022) [23]. GffCompare v. 0.11.2 (https://anaconda.org/bioconda/gffcompare; accessed on 1 August 2022) was used to compare the transcriptome data against the reference gene model [21]. The amino acids of the transcriptome data were compared against those for the reference gene model (GCF_003254395.2_Amel_HAv3.1_protein.faa; available in NCBI refseq) in BLAST v. 2.9.0+ (https://ftp.ncbi.nlm.nih.gov/blast/executables/blast+/LATEST/; accessed on 1 August 2022) using blastp with e-value <1 × 10^−10^ [24]. The signal peptide sequences for secretion were searched by SignalP v. 6.0g with Organism-Eukarya and fast mode [25].

### 2.3. Reference Transcriptome Sequence Annotations

The constructed transcriptome sequences were annotated via the functional annotation workflow Fanflow4Insects using the amino acid sequences predicted by TransDecoder (https://github.com/TransDecoder/TransDecoder; accessed on 1 August 2022) [18]. The protein-level functional annotation consisted of assigning the top hit genes from comprehensively annotated organisms using the sequence similarity search program in global alignment (GGSEARCH) (https://fasta.bioch.virginia.edu/; accessed on 1 August 2022) and annotating the protein domains using the protein domain database Pfam (http://pfam.xfam.org/; accessed on 1 August 2022) via HMMSCAN in the HMMER package (http://hmmer.org/; accessed on 1 August 2022).

### 2.4. Analysis of the Public RNA-Seq Data to Search for Novel Candidate Genes Involved in Immune Responses

Articles containing RNA-Seq data related to immune response in *A. mellifera* were searched in PubMed (https://pubmed.ncbi.nlm.nih.gov/; accessed on 1 August 2022) to retrieve immune-related RNA-Seq data for *A. mellifera* in public databases. The articles were curated, and SRA files related to the articles were downloaded and converted to FASTQ files. The FASTQ files were quality-controlled and trimmed via Trim Galore! v. 0.6.7 (https://github.com/FelixKrueger/TrimGalore, accessed on 1 August 2022). Transcripts per million (TPM) values for the reference transcriptome based on the selected RNA-Seq data were calculated with kallisto v. 0.46.0 (https://github.com/pachterlab/kallisto; accessed on 1 August 2022) [26].

To search for genes involved in immune response, the transcriptome scores for each experiment were calculated as shown in Equation (1) to avoid splitting by zero.
Score = {(average TPM values for pathogen-challenged samples) + 1} / {(average TPM values for control samples) + 1}(1)

Transcripts with scores >2 or <0.5 per experiment were listed. Novel candidate genes were searched by comparing lists with Calculate and plotting Venn diagrams (https://bioinformatics.psb.ugent.be/webtools/Venn/; accessed on 1 August 2022). For the gene set enrichment analyses, the transcripts were converted to human gene symbol annotations and used as the input for Metascape v. 3.5 (https://metascape.org/; accessed on 1 August 2022) [27].

## 3. Results

### 3.1. Construction of Reference Transcriptome of a Mellifera Using Public RNA-Seq Data

We curated all RNA-Seq data for *A. mellifera* in the GEO database to construct reference transcriptome sequences. From 50 available projects (~1300 runs as of April 2021), we selected the SRA data for 35 projects (1008 runs; Appendix A) after manual curation excluding duplicates, irregular conditions, and old types of NGS instruments. The curated RNA-Seq data were mapped to the genome data [3]. The reference transcriptome sequences were constructed using the mapped data and the reference *A. mellifera* gene model data. The transcriptome sequence contained 149,685 transcripts. There were 28,324 mRNAs in the gene model previously published in NCBI (GCF_003254395.2_Amel_HAv3.1_genomic.gtf) and designated RGM (reference gene model) (Figure 2A and Appendix A). There were 12,105 and 16,735 gene loci at the reference transcriptome and genome levels, respectively (Figure 2B and Appendix A). 11,993/12,105 loci of RGM (about 99%) match the loci of the reference transcriptome while 27,981/28,324 transcripts of RGM (about 99%) match the transcripts of the reference transcriptome. Thus, numerous gene isoforms predicted in RGM were identified in the reference transcriptome. Transcript IDs starting with “MSTRG” were newly identified in the present study. Taken together with these results, the reference transcriptome data include new isoforms belonging to genes found in RGM or transcripts belonging to the newly identified gene locus. Gtf file of the reference transcriptome in Appendix A from Stringtie shows which the transcripts belong to the gene locus (for details of the gtf file, visit Stringtie website. URL: http://ccb.jhu.edu/software/stringtie/index.shtml?t=manual; accessed on 23 September 2022). To evaluate the quality of the reference transcriptome data, we assessed the quality of the reference transcriptome data by BUSCO with hymenoptera_odb10 dataset in transcriptome data assessment mode with default settings, and for the comparison, BUSCO analysis using the reference genome data was also performed. The result of BUSCO of the reference transcriptome data is C: 99.6%[S: 3.0%,D: 96.6%],F: 0.2%,M: 0.2%,n: 5991 while the result of the reference genome data is C:97.7%[S: 97.6%,D: 0.1%],F: 0.3%,M: 2.0%,n: 5991 (C, S, D F, M and “n” indicate Complete, Single, Duplicated, Fragmented, Miss and total number of BUSCO core genes). The result of BUSCO showed that all most all core genes of Hymenoptera were included in the constructed reference transcriptome sequences as complete sequences and reference transcriptome data contains more core genes than the reference genome data, suggesting that the reference transcriptome data possesses comprehensiveness as omics data acceptable quality for reference data. The reason why most of the BUSCO assessment results are filled with D with C might be that the reference transcriptomes include many isoforms of the core single-copy genes.

We used the reference transcriptome sequences to predict the amino acid sequence with TransDecoder. Among 149,685 transcripts in the reference transcriptome, 194,174 amino acid sequences were predicted from 116,863 transcripts while the remaining 32,822 were noncoding transcripts (Appendix A). In RGM, 23,471 transcripts were coding transcripts among 28,324 transcripts (mRNAs) (Appendix A). The output file of TransDecoder showed that among the 116,685 coding transcripts, 49,003 transcripts (approximately 42% of the transcripts) coded multiple amino acid sequences, and 67,860 transcripts coded single amino acid sequences. We evaluated the predicted data and the amino acid sequences of RGM via blastp analyses (Appendix A). The comparison revealed that 154,466/194,174 (about 80%) of the reference transcriptome amino acid sequences hit the RGM sequences whereas 19,695/23,471 (about 84%) of the RGM amino acid sequences hit the reference transcriptome sequences of amino acids sequences with 100% identity and e_value = 0.0, and most of the remaining sequences showed over 99% identity and e_value = 0.0. Signal peptide sequences for secretion in the amino acid sequences were searched. The results showed that the 13,341 amino acid sequences have the signal peptide sequences among the 194,174 amino acid sequences of the reference transcriptome while the 2754 amino acid sequences of RGM have the signal peptide sequences among the 23,471 RGM amino acid sequences (Appendix A). We used Fanflow4Insects to conduct a functional annotation of the reference transcriptome using the amino acid sequences (Appendix A) [18]. Fanflow4Insects disclosed that ~62–47% of the amino acid sequences were functionally annotated via the *Homo sapiens*, *Mus musculus*, *and Caenorhabditis elegans* protein sequence datasets while ~62–50% were functionally annotated through the *Drosophila melanogaster*, *Bombyx mori*, *Bombus terrestis*, and *Nasonia vitripennis* protein sequence datasets (Table 1). We annotated 65.60% and 55.59% of the sequences with the UniGene and Pfam datasets, respectively.

### 3.2. Meta-Analysis of Reference Transcriptome and Published RNA-Seq Data to Detect new Candidate Genes (Transcripts) Involved in the Immune Response

To determine the utility of the reference transcriptome data, we performed a meta-analysis to search new candidate genes (transcripts) using the constructed and public RNA-Seq data related to the immune response. To retrieve the RNA-Seq data, we searched articles in PubMed using the words “apis mellifera immune rna seq”. We obtained 17 hits and curated them and their RNA-Seq data. We evaluated the article contents and the completeness and accessibility of the RNA-Seq dataset. After curation, we prepared the RNA-Seq data from five articles for use in the transcriptome expression analysis (Table 2). We prepared sequence data from the five bioprojects in the five articles. We calculated the TPM values of each RNA-Seq dataset after raw sequence data trimming and quality control. The TPM values of each experiment listed in Table 2 appear in Appendix A. We calculated the transcript scores for each experiment sought to identify those that were upregulated or downregulated in response to pathogen infection (Appendix A). Those that were upregulated and had transcript IDs with scores > 2 were listed in each experiment (Appendix A). No single upregulated transcript was common to all experiments. Thus, we searched for upregulated transcripts in the experiments related to bacterial infection (AFB, B_BN, P_PN, M_MN, and C_CN) or viral infection (CC_CV, RC_RV, SBV_DWV, and DWV). Figure 3A,B show that 20 and 10 transcripts were upregulated in the viral- and bacterial infection-related experiments, respectively. The emergent transcript IDs in each category are listed in Appendix A. We also searched downregulated transcripts with scores <0.5. No single downregulated transcript was common to all experiments (Appendix A). Nine and three transcripts were downregulated in all viral infection- and bacterial infection-related experiments, respectively (Figure 3C,D; Appendix A).

We used the annotation data to examine the transcripts upregulated or downregulated in response to viral or bacterial infection (Appendix A). We used Metascape to perform an enrichment analysis on the 20 transcripts upregulated in response to viral infection. The inputs were annotations of human gene symbols converted by the annotation data. In the gene ontology (GO), the transcripts related to cellular protein catabolic process (GO: 0044257) or protein ubiquitination (GO: 00165567) were included in the 20 transcripts (Appendix A). However, *p* values for both ontologies were not low. The transcripts upregulated in response to bacterial or viral infection did not include immune-related IMD, Toll, or JAK-STAT pathway or autophagy (Appendix A) [33,34]. However, aldehyde dehydrogenase 3 (XM_026442860.1) and negative regulator of RNA polymerase III (MSTRG.11647.15) were included among the transcripts upregulated in response to bacterial infection. In contrast, *autophagy-related protein 3* (*ATG3*) (MSTRG.1676.15) was included among the transcripts downregulated in response to viral infection while urea transporter transcript (MSTRG.12160.40) was downregulated in response to bacterial infection.

## 4. Discussion

The present study was a meta-analysis intended to provide reference transcriptome data for the honeybee *Apis mellifera*. We constructed the *A. mellifera* reference transcriptome sequence data using RNA-Seq data curated from the public database and reference chromosome-level genome sequence and their gene set data [3]. Fanflow4Insects was used for functional transcriptome annotation [18] integrating model species such as humans, *M. musculus*, *C. elegans*, and several insect species such as *B. mori*. We then performed another meta-analysis using reference transcriptome data constructed and public RNA-Seq data of immune-challenged *A. mellifera* to search novel candidate genes related to the immune response. We identified several candidate transcripts.

The reference transcriptome sequence consisted of 149,685 transcripts belonging to 16,735 loci. Whereas the RGM had 28,324 transcripts and 12,105 loci, the reference transcriptome had far greater numbers of both. Large numbers of RNA-Seq data from the public database were used as hint data to construct the reference transcriptome sequence. The public RNA-Seq data were derived from neurons, midguts, hypopharyngeal gland mushroom bodies, and other tissue samples at multiple developmental stages such as the embryo and the pupa (Appendix A). The abundant RNA-Seq data might contain the transcripts or isoforms expressed in the various tissues and developmental stages and possessing important biological functions. In mosquitoes, Down syndrome cell adhesion molecule (Dscam) mediates phagocytosis and is a pattern recognition receptor. It has many isoforms and is divergently associated with isoform-specific immune responses to various pathogens [35]. Thus, reference transcriptome data can reveal biological reactions related to isoform-specific functions. Moreover, the reference transcriptome includes loci previously unidentified in RGMs. We assumed that the locus transcripts were expressed at very low levels and at only a few developmental stages. The foregoing information suggests that transcriptome analyses using the reference transcriptome can accurately indicate entire expression states and disclose transcripts with isoform- and tissue-specific functions.

Functional transcriptome sequence annotations are critical in the biological interpretation of analytical results. For the functional annotation in the present study, 194,174 amino acid sequences from 116,863/149,685 transcripts were predicted because multiple amino acid sequences were predicted from a single transcript. Blastp against the RGM protein dataset demonstrated that the predicted amino acid sequences of the reference transcriptome covered nearly all the RGM protein data comprising 23,471 sequences. There were also 9935 coding genes in the Refseq data (Amel_HAv3,1: NCBI Apis mellifera Annotation Release 104). The foregoing findings suggest that the reference transcriptome data covered nearly all the amino acid sequences for the genes and transcripts in the previous version and also included novel heretofore unidentified genes and transcripts. Fanflow4Insects [18] was used to search transcriptome orthologs among the model and insects species. There were hits between ~60% of the amino acid sequences of the transcriptome sequences and the human and *M. musculus* gene sets. There were also hits between 50% of them and the *D. melanogaster* gene set. A possible explanation is that the human and *M. musculus* gene sets were more sophisticated and better annotated than those of the other species. As more knowledge regarding these species is gained in the future, their annotations will improve. The functions of several amino acids were revealed but these data were not “functionally annotated” or incorporated into the public database. Several predicted amino acid sequences belonged to a single transcript. Usually, one matured mRNA encodes one amino acid. Therefore, some amino acid sequences from one transcript sequence may be incorrectly predicted. Hence, such sequences cannot be functionally annotated, and the total percentage of hits might be reduced. The annotation results may be used in transcriptome enrichment analyses of GO and Kyoto Encyclopedia of Genes and Genomes (KEGG) annotations by converting the reference transcriptome ID to the gene symbol for the model species in the annotation file. Our amino acid sequence and annotation data may facilitate biological interpretations of the results of the transcriptome analyses.

We used reference transcriptome and RNA-Seq data from the public database to perform a meta-analysis and detect novel candidate genes (transcripts) related to the immune responses to viruses or bacteria [28,29,30,31,32]. Pathogens infecting honeybees are economically serious problems, bringing about mass deaths of honeybees [1]. One of the resolutions to the problems is the enhancement of immune reactions of honeybees through gene modifications such as genome editing. Thus, searching for novel candidate genes related to the immune responses and understanding the immune system in honeybees can lead to the first step of the resolution. Therefore, we searched for the novel genes. We calculated the expression levels of the transcriptomes and the RNA-Seq data as well as the changes in transcriptome expression in response to viral or bacterial infection. We applied the scores for the published immunological experiments to search the downregulated and upregulated transcripts common to them all. However, we found none. We then searched the upregulated or downregulated transcripts common to four viral or five bacterial experiments. There were 20 upregulated and 9 downregulated transcripts among the viral experiments, and 10 upregulated and 3 downregulated transcripts among the bacterial experiments. Hence, there were very few transcripts and none of them were functionally annotated as immune-related. Possible explanations for these discrepancies were the facts that the experiments widely varied with respect to incubation time, *A. mellifera* developmental stage, inoculation procedure and difference of beekeeping styles between laboratories. In addition, *A. mellifera* samples are usually prepared from different colonies with diverse genetic backgrounds, and are therefore genetically diverse (meaning genetic variations in individual samples). These factors could affect gene expressions. The TPM values of the identified transcripts widely varied. Nevertheless, the candidate transcripts could participate in the immune response as they emerged under highly diverse conditions. The reasons for not including immune-related genes, such as the genes consisting of Toll or IMD pathway, in bacterial up-regulated genes in meta-analysis might be that the bacterial-related RNA-Seq data from bees or larvae post “orally infection” 3 or 7 days were used. In *Tribolium castanaum*, the inductions of the most immune-related genes peaked post bacterial injections 6h while some of them peaked 24 h [36]. Furthermore, generally, intensities of immune gene inductions by the “injections” of bacteria are higher than the “ingestions”. Therefore, these immune-related genes could be filtered out or not induced in the meta-analysis of RNA-Seq data we used in this study. *ATG3* was included among the common transcripts downregulated in response to viral infection and may be implicated in ubiquitin-like conjugation systems that prolong the autophagosomal membrane phase. The autophagosomal membrane phase is one of three autophagy processing stages. *ATG3* functions in *D. melanogaster* [37]. Autophagy suppresses viral replication. When autophagy-related genes were silenced in *D. melanogaster* challenged with vesicular stomatitis virus (VSV), the insects presented with higher mortality rates than the control. We assumed that in *A. mellifera*, *ATG3* was downregulated in response to viral infection to avoid the host immune response and autophagy. When Toll-7 detected VSV in *Drosophila*, autophagy activation did not depend on canonical immune signaling pathways such as IMD or Toll [37]. This finding was consistent with that of our meta-analysis, which detected no immune-related transcripts among those commonly upregulated or downregulated in response to microbial infections. The foregoing results suggest that autophagy was implicated in the viral immune responses in *A. mellifera* and autophagy regulation regardless of the canonical immune pathways are conserved in both *D. melanogaster and A. mellifera*. In future work, meta-analysis using small numbers of RNA-Seq data in fewer studies, more relevant RNA-Seq data, or RNA-Seq data using pure ligands such as peptidoglycan or lipopolysaccharide could reveal new immune-related genes.

Half of the immune-related-candidate genes (transcripts) were not included in RGM (IDs of them start “MSTRG”.). The reason for it can be that the reference transcriptome includes many transcripts which are not included in RGM. As described above, the reference transcriptome data were constructed using many RNA-Seq data as hint data derived from multiple tissues, developmental stages, different castes, and the bees under many conditions (e.g., infecting bacteria), leading to identifications of the new transcripts. Even though several false positive transcripts could be included in the transcriptome data, we consider that the important point of the reference transcriptome is comprehensiveness: the transcriptome data include the transcripts expressed only in bees under specific conditions, at specific developmental stages, or in specific tissues. In addition, we performed annotations of the transcriptome sequences using gene sets of model species (e.g., human and mouse) and several model insect species, resulting in ~60 % transcriptome sequences annotation [18]. Consequently, *ATG3* transcript described in the previous paragraph was newly identified in the reference transcriptome. The “*ATG3*” annotation was derived from human gene sets. Considering them, transcriptome analysis using either public RNA-Seq data or newly sequenced RNA-Seq data and the reference transcriptome data could lead to new biological insights and interpretations of honeybees.

In the present study, we constructed reference transcriptome data and showed that they could readily be applied towards a meta-analysis. The results of the latter showed that transcriptome expression widely varied in response to microbial infection in *A. mellifera*. In future studies, RNA-Seq data of *A. mellifera* under strictly controlled conditions and at all developmental stages will be needed. In this manner, the dynamics of transcriptome expression in *A. mellifera* may be elucidated. The transcriptome data could also be used in these RNA-Seq data analyses or RNA-Seq analyses of conditioned samples such as pathogen-infected worker bees. In addition, comparative transcriptome analysis between castes using the reference transcriptome data and RNA-Seq of each caste could reveal the molecular mechanism of the differences in the biological features among castes, such as differences in immune responses and morphological, behavioral, and physiological differences. Since the transcriptome constructed by RNA-Seq of diverse honeybees derived from multiple castes can include these isoforms, our reference transcriptome could contribute to the identifications of caste-specific isoforms involved in the differences among castes. These investigations will help clarify the molecular biology of *A. mellifera*.

## Figures and Tables

**Figure 1 insects-13-00931-f001:**
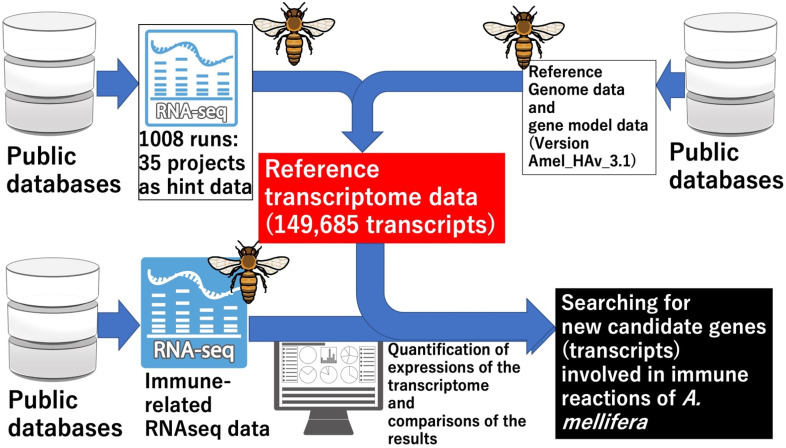
Workflow of the data analyses performed in the present study. To construct the reference transcriptome data, 1008 RNA-Seq data from 35 projects were mapped to the reference genome data used as hint data. The reference transcriptome data were constructed using a combination of reference genome, genome model, and hint data and they were functionally annotated with Fanflow4Insects [18]. We retrieved and curated immune-response-related RNA-Seq data in the public database to search novel candidate transcripts implicated in these processes. We quantified the expression levels of the RNA-Seq data transcripts using the reference transcriptome data. We searched for new candidate transcripts by comparing the expression data. Images were obtained from TogoTV (© 2022 DBCLS TogoTV).

**Figure 2 insects-13-00931-f002:**
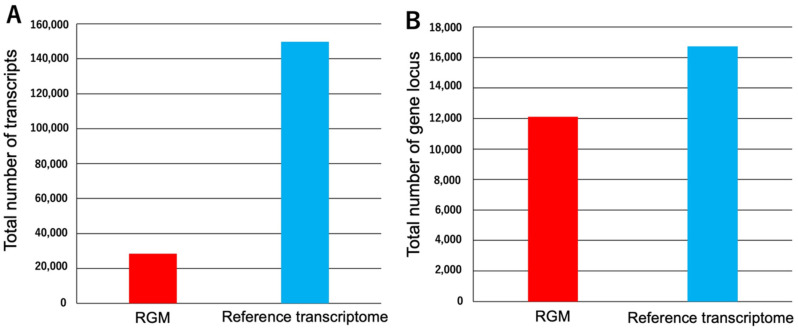
Comparisons of the total numbers of transcripts and gene loci in the reference gene model (RGM) and the reference transcriptome data. (**A**) Comparisons of the total number of transcripts and both datasets. (**B**) Comparisons of the total numbers of gene loci in both datasets via the gffcompare program.

**Figure 3 insects-13-00931-f003:**
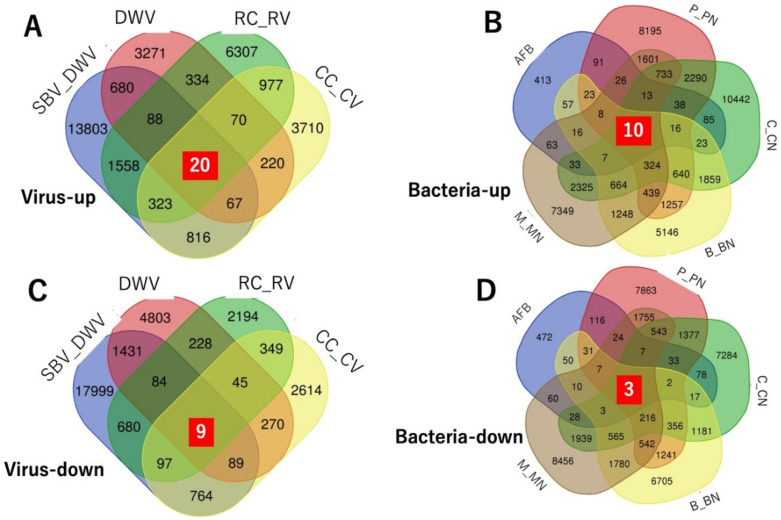
Transcripts upregulated or downregulated in response to viral or bacterial infection. Numbers in Venn diagrams indicate numbers of upregulated transcripts in experiments (Table 1) on viral infection (**A**) or bacterial infection (**B**) and numbers of downregulated transcripts in experiments on viral infection (**C**) or bacterial infection (**D**). White numbers in red squares indicate numbers of upregulated or downregulated transcripts in all experiments in (**A**–**D**).

**Table 1 insects-13-00931-t001:** Number of annotated amino acid sequences.

Protein Sequence Dataset	*H. Sapiens*	*M. Musculus*	*C. Elegans*	*D. Melanogaster*	*B. Mori*	*B. Terrestris*	*N. Vitripennis*	UniGene	Pfam
No. annotated predicted amino acid sequences (total 194,174)	121,027	115,617	91,859	97,544	121,119	113,306	110,591	127,386	107,936
% annotated sequences	62.33%	59.54%	47.30%	50.24%	62.38%	58.35%	56.95%	65.60%	55.59%

**Table 2 insects-13-00931-t002:** Public RNA-Seq data used in expression calculations.

Bioproject	Sample	Treatments	Reference	Sequence Read Archive Accession ID (Sample (Abbreviation)	Experimental Abbreviation)
PRJNA52851	4th instar honeybee larvae	*Paenibacillus larvae* orally inoculated and treated larvae were incubated 72 h.	[28]	SRR068395 (infected), SRR068396 (Control).	AFB
PRJEB6511	Newly hatched larvae	Sacbrood virus (SBV)-deformed wing virus (DWV) mixture orally inoculated and treated larvae were incubated 9 d. Control was PBS.	[29]	ERR528750 (control_1_D1), ERR528751 (control_2_D2), ERR528753 (DWV_SBV_1_C5), ERR528754 (DWV_SBV_2_C6),	SBV_DWV
PRJNA445764	Abdomens of worker adults	1. Diet, Pollen (P), Bee-Pro (B), MagaBee (M) or control (carbohydrate diet) (C) fed 7 d. 2. *Nosema apis* and sucrose provided. 3. Each diet fed 7 d.	[30]	SRR6901841 (B2), SRR6901842 (B1), SRR6901848 (B3), SRR6901856 (BN1), SRR6901857 (BN2), SRR6901858 (BN3) SRR6901843 (P1), SRR6901844 (P2), SRR6901849 (P3), SRR6901859 (PN1), SRR6901860 (PN2), SRR6901862 (PN3) SRR6901845 (M1), SRR6901846 (M2), SRR6901847 (M3), SRR6901853 (MN1) SRR6901854 (MN2) SRR6901855 (MN3) SRR6901850 (C1) SRR6901851 (C2) SRR6901852 (C3) SRR6901861 (CN1) SRR6901863 (CN2) SRR6901864 (CN3)	B_BN (control vs. *N. apis*-inoculated samples reared with Bee-Pro) P_PN (control vs. *N. apis*-inoculated samples reared with pollen) M_MN (control vs. *N. apis*-inoculated samples reared with MagaBee) C_CN (control vs. *N. apis*-inoculated samples reared with carbohydrate diet)
PRJNA498919	Whole bodies of adult bees	Samples reared with chestnut or rockrose (less nutritious), fed control or inoculated with Israeli acute paralysis virus (IAPV) containing 30% sucrose solution for 36 h	[31]	SRR8121077, SRR8121078 (CC_1), SRR8121079, SRR8121080 (CC_2), SRR8121081, SRR8121082 (CC_3), SRR8121083, SRR8121084 (CC_4), SRR8121085, SRR8121086 (CC_5), SRR8121087, SRR8121088 (CC_6), SRR8121101, SRR8121102 (CV_1), SRR8121103, SRR8121104 (CV_2), SRR8121105, SRR8121106 (CV_3), SRR8121107, SRR8121108 (CV_4), SRR8121109, SRR8121110 (CV_5), SRR8121111, SRR8121112 (CV_6) SRR8121089, SRR8121090 (RC_1), SRR8121091, SRR8121092 (RC_2), SRR8121093, SRR8121094 (RC_3), SRR8121095, SRR8121096 (RC_4), SRR8121097, SRR8121098 (RC_5), SRR8121099, SRR8121100 (RC_6), SRR8121113, SRR8121114 (RV_1), SRR8121115, SRR8121116 (RV_2), SRR8121117, SRR8121118 (RV_3), SRR8121119, SRR8121120 (RV_4), SRR8121121, SRR8121122 (RV_5), SRR8121123, SRR8121124 (RV_6)	CC_CV (control vs. IAPV-inoculated samples reared with chestnut) * RC_RV (control vs. IAPV-inoculated samples reared with rockrose) *
RJNA669279	Larvae	DWV containing PBS (or OBS as control) were inoculated 12 h in larvae reared for 3 d from 1-d larvae. Treated larvae were incubated 4 d.	[32]	SRR12830831 (control_1), SRR14424854 (control_2), SRR14424852 (control_3), SRR12830832 (DWV_1), SRR12830855 (DWV_2), SRR12830853 (DWV_3)	DWV (control vs. DWV-infected)

* Two run files were tied to a single sample and merged into one file. TPMs were calculated using merged data.

## Data Availability

All Appendix A are available in figshare (DOI: 10.6084/m9.figshare.c.6047651).

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
