# Peer review of "Meta-Analysis of the Public RNA-Seq Data of the Western Honeybee *Apis mellifera* to Construct Reference Transcriptome Data"

_insects, 2022, doi:10.3390/insects13100931_

Round 1

Reviewer 1 Report

The authors newly constructed reference transcriptome of A. mellifera, and succeeded in finding several novel candidate genes possibly involved in immune response. I think the topic is important and this study contains significant findings, although some points should be modified before publication from Insects.

Major points

1. To test the utility of the reference transcriptome, the authors performed the meta-analysis to identify novel candidate genes involved in the immune response. 

              a) Honeybee show a wide variety of unique biological phenomena. Therefore, please clarify why the authors specifically focused on the immune response and the importance of understanding the immune system in honeybee biology. 

              b) As shown in Supplemental data 10, about half of the up- or down-regulated transcripts have IDs starting with "MSTRG", supporting the utility of the reference transcriptome. I recommend the authors clearly describe this positive result and discuss the potential of the new reference transcriptome data. In addition, one of the most important finding in this study is identification of ATG3 as down-regulated genes in response to the viral infection. The transcript annotated as ATG3 is also MSTRG~. Was this gene not predicted in "RGM"? If so, why?

2. The society of honeybee consists of the workers, a queen, and drones. In my opinion, to uncover the molecular mechanism underlying the morphological, behavioral, and physiological difference among these types of bees by comparative approach is fundamental topic in the field of honeybee biology. I agree that the meta-analysis would be a powerful tool in future study. The authors, however, discuss future direction on the meta-analysis in honeybee only focusing on the workers. Please describe how to contribute the reference transcriptome constructed in this study (and limit, too) from broader perspective of honeybee biology.

Minor points

1. I feel that the contents of Simple Summary and Abstract are substantially identical. I think that the role of Simple Summary is appealing to public. I suggest the authors rewrite this part with emphasis on "the importance of honeybee", "the needs of comprehensive transcriptome data", and "the significance of the findings". 

2. Table 1. 3rd column: "% antedated sequences" --> "% annotated sequences"?

3. ls.282–283: "~ transcripts were predicted as multiple amino acid sequences were predicted from a single transcript." Grammatically correct?

4. ls.320–322: "A. mellifera samples were genetically diverse." 

What "genetically" means? Is there the difference of race (subspecies), such as Italian, Caucasian, and Carniolan, among the studies which the authors selected? I agree that difference of beekeeping styles between laboratories could result in the inconsistency between the studies. However, this is physiological difference rather than genetical difference. 

5. ls.327 & 335: I cannot find what the authors intend to describe using "the latter".

6. l.331: "assumed that In A. mellifera" --> "assumed that in A. mellifera"

7. l.338: "pathways conserved in" --> "pathways are conserved in"

Reviewer 2 Report

The manuscript presented by Yokoi, et al described the meta-analysis of around 1000 RNA-seq runs for Apis mellifera and constructed new reference transcriptomes. Using the constructed reference transcriptomes, they identified 3-20 transcripts that were regulated in response to viral or bacterial infection. Generally, this study provides insight into the construction of new reference transcriptomes for A. mellifera and identifying new immune-related transcripts. However, the authors were not able to identify known immune-related transcripts using the transcriptomes they constructed, which makes it challenging whether the transcriptomes need to optimize or how to better use this database. There are some comments:

1. The authors identified 3-20 transcripts regulated in response to viral or bacterial infection. However, none of them was annotated as immune-related. The authors gave the explanations that the exp included in this study widely varied and the A. mellifera samples were genetically diverse. I think another reason might be the challenges were quite different. Though the authors limited the pathogens to viruses and bacteria, these pathogens are very different, they may cause totally different immune responses. In this case, it's not surprising that not many shared transcripts were identified. I would suggest the authors choose fewer studies to do the analysis or take more relevant data to combine the results. Another strategy the authors can try is instead of pathogens challenging, it’s easier to analyze pure ligand stimulation data, such as LPS stimulation.  

2. The authors made a good point that the A. mellifera samples might be genetically diverse, as they are not under strictly controlled conditions. A. mellifera is not a widely used model organism, and its gene expression profile can be altered by different culture conditions or reproductive strategies. This increases the difficulty of building a unified transcriptome database. This somewhat diminishes the importance of this study. However, this study still provides a new means to analyze the transcriptome data of insects. This may have application value in the future.

Reviewer 3 Report

Yokoi et al. described in this manuscript an analysis of the 1008 RNA-seq datasets collected from the honeybee, Apis mellifera, and an assembly of the 149,685 transcripts and 194,174 amino acid sequences. They identified 320 transcripts responsive to both viral and bacterial infection and found ATG3 was down regulated upon viral infection.

While the transcriptome is certainly useful, the authors have not clearly demonstrated the expanded sequence lists represent a substantial improvement of Amel_Hav_3.1. A comparison is needed to either support some of the gene models in Amel_Hav_3.1 or provide alternative ways to splice the initial gene transcripts.

Even if the 149,685 transcripts do not include long noncoding RNA, is it true that 30% of the transcripts have more than one translational initiation site or being bi-cistronic? Perhaps, a considerable part of the transcript and protein sequences represent false positive.

On the other hand, identification of the 10 upregulated transcripts after bacterial challenge (<20 after viral infection?) may be a result of severe false negatives. Since induction of immunity-related genes is well established in A. mellifera, this artifact could be a problem of pooling many RNA-seq datasets collected under various conditions by different peoples, as discussed by the authors.

Consequently, l would like to suggest the authors to consider these fatal issues and try to improve their study from scratch. To improve the work, the authors may consider learning a method that uses less transcriptome sequence information to generate reliable gene models (Cao and Jiang, 2015, Integrated modeling of protein-coding genes in the Manduca sexta genome using RNA-Seq data from the biochemical model insect. Insect Biochem. Mol. Biol. 62, 2−10.)

I look forward to seeing a drastically improved manuscript.

Round 2

Reviewer 3 Report

In the revised manuscript, Yokoi addressed some of the concerns of this reviewer.

Response 1: Results from a DIRECT comparison with all gene models is needed to support this is a substantial improvement of Amel_Hav_3.1, including 1) ORF length identity, difference, and ratio; 2) gene model redundancy; 3) coding and noncoding transcripts; 4) presence or absence of complete signal peptide for secretion; 5) examples for 1) through 4).

Response 2: When number of datasets reached, say, 100, a further increase to 1000 (1008 in this case) is going to help but never cover the entire transcriptome, especially for the rare ones. Arguments about more isoforms in specific stages or tissues needs SOLID data support. Please provide that. After noncoding transcripts and transcripts with more than one translational initiation site are removed from the total of 149,685, what percentage of multi-cistronic genes is in the remaining part? Is it plausible?

Response 3: I accept the authors’ explanations on immunity-related transcripts in general.

Response 4: It is understandable that the authors do not follow a specific protocol suggested by a reviewer. However, in this case, it may be of use when authors try to address the concerns in Response 1. By the way, BUSCO scores have limitations. What are the BUSCO scores of Amel_Hav_3.1? Are there substantial increases? By the way, Fig. 3 is not needed: reporting the scores in the text is sufficient.

Round 3

Reviewer 3 Report

There are small glitches in the paper, making it hard to read. Please carefully examine the proof to ensure there is no broken sentences or other grammatical errors.